# Ideas and perspectives: Ushering Indian Ocean into the UN Decade of Ocean Science for Sustainable Development (UNDOSSD) through Marine Ecosystem Research and Operational Services – An early-career's take

Kumar Nimit

Indian National Centre for Ocean Information Services (INCOIS), Ministry of Earth Sciences (MoES), Hyderabad-500090, India

*Correspondence to*: nimitkumar.j@incois.gov.in

**Abstract.**

The Indian Ocean-rim hosts many of the world's underdeveloped and emerging economies that depend on the ocean resources for the livelihoods of the populations of millions residing in. Operational ocean information services cater to the requirements of managers and end-users to efficiently harness those resources, and to ensure safety. Fishery information is not the only operational services that will be needed to empower such communities in the coming decades. Coral bleaching alerts, SCUBA-assist advisories, conservation or ecotourism assist services (e.g. Turtle-watch/Whale-watch), poaching/bycatch reduction support, jellyfish/microplastic/oil-spill watch are to name a few, but not an exhaustive list of the services that are needed operationally. This paper outlines existing tools, and explores the ongoing research that has potential to convert the findings into operational services in near-to-mid term.

**Keywords:**

Ecosystem, IIOE-2, Operational Services, UN Decade of Ocean, Perspectives

## 1. Introduction

About a third of the world population reside in the Indian Ocean -rim (IOR) countries. This is projected to increase and may reach to 50% of the global populations by the year 2050 (Doyle, 2018); whose lives and livelihoods are inherently intertwined with the Indian Ocean and its dynamics. These countries have very diverse cultures and philosophies. Their financial capabilities vary widely and so do their aspirations and world-views. The IOR countries thus present a confluence of ideologies to a manager, with their vivid and vibrant demographics that have very different challenges and opportunities. Majority of the IOR countries have either expansive or stationary population pyramid profile, with limited land resources. Their nutritional and employment needs will have more reliance on the Indian Ocean in coming decades. Marine ecosystems are a promising source for the nourishment of populations and at the same time, are under more serious threats than ever before. As more people are brought up from the poverty, consumerism is poised to extend to larger parts of the population, which will create concerns for the health of the adjoining ocean ecosystems.

The International Indian Ocean Expedition (IIOE) conducted during 1962-1965, was the first major attempt to study the Indian Ocean, its processes and diversity. It resulted into many multidisciplinary collaborations and gave birth to institutions such as National Institute of Oceanography (NIO) in India. The second IIOE (IIOE-2) was observed during 2015-20 commemorating five decades of the first IIOE, and witnessed even greater participation globally. It has been underlined on the onset IIOE-2 that, the Indian Ocean is one of the relatively poorly studied oceans of the world (Hood et al., 2016). It is thus important to ensure that the resources from this unique (land-locked from the north) ocean are harnessed in harmony with sustainability, and with the equal opportunities. For the successful planning and execution, it is imperative for the resource managers to have community participation (Martini et al., 2017). Hence, it is especially important for the IOR countries to chart the right path at the eve of the United Nations Decade of Ocean Science for Sustainable Development (UNDOSSD, 2021-2030).

Advancement of geosciences in last century has provided us many tools to cope with the changing environment and associated challenges. The role of providing such insights in the twenty first century is shifting to operational oceanography (OSOS, 2019). Operational oceanography is defined as systematic and sustained ocean observations which are converted into outputs as per the stakeholder/user requirements; using the scientific research done in the development of those outputs. Operational ocean services are the mechanism through which these outputs are disseminated effectively (often, in real-time or near real-time) and feedbacks are collected on users' experience and needs (Fig.01). Robust combination of observations and modelling, artificial intelligence, effective tools of interpretation and user-friendly dissemination of information can change millions of lives dependent on the oceans. This paper hypothesizes that the best possible approach to win the community attention (and to invoke the responsible ownership of the resources) is to engage the communities with the operational ecosystem related services. Further, this paper explores the regional needs and solutions based on presently available data and technologies.

**2. Marine Ecosystem Research and operational Services (MERS)**

The marine ecosystem related operational ocean information and forecasts services (or products as a service) can be grouped into three major categories based on their prime objectives. The first type of services is related to resource management, conservation and support to livelihood. The second type comprises of services that deal with threats to the ecosystem. The third group is of the solutions for community engagement and dissemination.

2.1 Services to cater resource management, conservation and livelihood

Under the decadal initiative, the UN Sustainable Development Goal (SDG)-14 (Life below water) will receive significant attention, which heavily relies on remote sensing data in general and in particular, on Ocean Color (OC) observations. In the late 90's, the OC science achieved operational status (after a decade long hiatus post-CZCS era) due to adequate satellite coverage. Since then researchers have made extensive use of these data despite often encountering spatial (clouds and sun-glint), temporal (daily once, daytime only coverage) or spectral (fixed, limited bands) limitations. The applications range wide from fisheries and aquaculture to Harmful Algal Bloom monitoring to estimating productivity and carbon budgeting (Wilson, C., 2011). However, there are limitations with regard to predictability of the base of the food pyramid itself. There are global products for primary production such as Vertically Generalized Productivity Model that are developed with relatively low observations of the Photosynthesis-Irradiance parameters from the Indian Ocean. It is thus very important for the IOR countries to have regional Primary Productivity models for operational applications. This could be taken up as a priority during the UNDOSSD.

This decade is chiefly important due to the planned launch of an operational hyperspectral mission named PACE (Plankton, Aerosol, Cloud, ocean Ecosystem), by NASA (National Aeronautics and Space Administration, USA) in the early 2020's. A hyperspectral sensor scans oceans with high-resolution of as much as 1nm wavelength for the entire visible spectrum, that enables the researchers to begin a new era in the OC sciences due to its potential of developing a plethora of applications. This includes but is not limited to better resolving phytoplankton functional types, fishery resource management and ecosystem health monitoring. The latter is more important especially for the second type of services and is discussed in detail in a subsequent section.

At the same time, the temporal limitations seem to be mitigated through various approaches such as using the Day-Night Band data or OC data from the geostationary orbit such as Geostationary Ocean Color Imager (Miller et al., 2013; Ruddick et al., 2014). When OC science follows the footsteps of other satellite types, we may also have a "swarm" of nano-satellites or cubesats – further improving temporal and up to some extent, spatial coverage as well. This however needs to be rightly balanced with the concerns of overcrowding the orbits and associated complexities (Bruinsma et al., 2021). Studies are required to shed light on the fate of ocean observations platforms (chiefly, drifting platforms) and setting up best-practices, to minimize their contribution to marine debris. Placement of HICO (Hyperspectral Imager for Coastal Ocean) on International Space Station (ISS) reminds us that the OC science may not be limited to satellite remote sensing in coming decades (Huemmrich et al., 2017). Validations, sampling protocols, inter-comparison exercises and training for all of

these will be needed. Modelers are putting efforts to assimilate data products into numerical models or, making
use of surface information for better resolving subsurface characteristics. Others strive for carbon, or heat-
budgeting. With more than two decades of uninterrupted observations, (applying the rule of thumb of minimum
30 years' time-series) we are a few years away of making use of OC data for climate applications.
There are some proven (e.g. telemetry) and emerging (e.g. eDNA) tools that can help researchers for the
services related to higher trophic levels (Costa et al., 2010; Stat et al., 2017). These tools are very useful in
providing vital information habitat utilization, migration, environmental preferences and presence-absence
analysis. The research can then support the operational services that cater to sustainable resource management,
prevention of bycatch or help reduce accidental mortality. TurtleWatch advisory by NOAA (National Oceanic
and Atmospheric Administration, USA), is the best suited example for the majority of the Indian Ocean
countries (Howell et al., 2015). Other species for which such products are very much needed in the operational
mode are Cetaceans, Sharks and marine mammals.
The operational services that can predict the movement and presence of these animals are required for
conservation authorities who can then effectively design protected areas, do better marine spatial planning or
monitor potential hotspots for poaching or IUU (Illegal, Unreported and Unregulated) fishing. Additionally
these can also support the ecotourism (e.g. WhaleWatch) as an alternate livelihood for the fishermen that face
capture fishery as a less lucrative occupation. Ecotourism needs support from operational services monitoring of
mangroves and corals as well. Innovative ideas that put together existing observations and forecasts (e.g. water
clarity from OC satellite data, currents and wave-height forecasts from numerical model simulations, and coral
health i.e., no-bleaching/stress alert with the help of satellite-based SST), and generate novel products (e.g.,
SCUBA-buddy advisories), will support dive operators in many of the IOR-countries, especially the SIDS
(Small Island Developing States). Mariculture (open ocean cage culture) is emerging as another area of
livelihood as an alternate to the capture fishery and is poised to be more feasible at wider areas with newer
designs of submerged cages (Korsøen et al., 2012).
Acoustics is another tool that can support development of operational services for various stakeholders.
Acoustic telemetry is useful for the species to which satellite telemetry isn't suitable due to various reasons
(Crossin et al., 2017), whereas the field of bioacoustics has been furthered into many sub-branches. The
estimation of zooplankton biomass with the help of acoustic backscatter isn't only limited to moored buoys and
these techniques have transcended to AUVs (Autonomous Underwater Vehicles), aptly called zooglider (Ohman
et al., 2019). Such surveys are vital for zooplankton and ichthyoplankton distribution monitoring, and for
ecosystem models that would connect the regional PP products to the higher trophic levels. Bioacoustics is also
used for monitoring the marine animals' movements by listening to the underwater sounds, often through
Passive Acoustic Monitoring (PAM), which would be one of the less intrusive ways of study (Brwoning et al.,

130 2017).

2.2 Services to mitigate threats to the ecosystem
Many of the human activities can be nuisance to the marine environment in different ways and anthropogenic
sound in the ocean is one such example (Williams et al., 2015). Future operational services will need to account
for and develop ways to monitor the oceanic soundscape.
Another common anthropogenic impact on the marine ecosystems is transport and introduction of alien (non-
native) species through the ballast water. These species when lack natural predators, are often turn out to be
invasive species and throw off the local ecological equilibrium (Bailey S.A., 2015). Oil spill is yet another
source of pollution from the ships. These can be addressed up to certain extent by monitoring vessel AIS
(Automatic Identification System) data and applying artificial intelligence-machine learning (AI-ML) tools to
identify the ship behavioural patterns prior to such discharges; and then use it as predictive model (Soares et al.,

141    2019).

Plastic adversely impacts the marine environment in multiple ways. The known pathways include ghost fishing
(entanglement), ingestion, bioaccumulation and allied effects, as well as transport of alien species and
pollutants. Among the top-20 countries globally with inadequately managed plastic as a source to the ocean, half
are the IOR countries. All of the Bay of Bengal peripheral countries face the issue of land-plastic management
(Jambeck et al., 2015). Thus, the Bay of Bengal needs to be a prime focus to monitor the oceanic and beach
debris in the mission mode, with the triad of *in-situ* surveys, dispersion models and latest remote sensing
techniques (Martínez-Vicente et al., 2019). Whether it is about addressing entry of the plastic from land to
ocean, or sustainable exploitation of shared fish stocks; progressive and tangible engagement among
stakeholders through forums such as SAARC (South Asian Association for Regional Cooperation), BIMSTEC
(Bay of Bengal Initiative for Multi-Sectoral Technical and Economic Cooperation), WIOMSA (Western Indian
Ocean Marine Science Association) or RIMES (Regional Integrated Multi-Hazard Early Warning System for
Africa and Asia), is a necessity.
On the other hand, mass sink of jellyfish swarming has been reported in the Arabian Sea (Billet et al., 2006).
Jellyfish swarms are known to be disruptive to fisheries and to the cooling water intake facilities of coastal
industries with causes little known, and thus the need for development of predictive capabilities. The Arabian
Sea also witnesses large scale blooms of *Noctiluca scintillans* and research in recent years may provide the
foundation for the bloom forecasting (Xiang et al., 2019). Crashing of jellyfish swarms or algal blooms have
implications to the deep sea ecology, particularly in regions such as the Arabian Sea, which has prominent
oxygen minimum zone (OMZ) (Rixen et al., 2020). .Stratified waters such as those of the northern Indian Ocean
basins have implications for large pelagic, which may face additional fishing pressure with the expansion of the
OMZs due to shrinkage of the habitat (Nimit et al., 2020).
2.3 Effective dissemination and user-engagement
While it is important to have robust observations and modelling capabilities in order to generate operational
services and products, it is equally important to have effective dissemination. Presently, the IOR countries rely
on information dissemination through conventional modes that are almost entirely land-based. Dissemination of
operational services onboard has its own benefits during fair weather as well in terms of fuel saving. However,
due to limitation of cellular signal support at sea, it is imperative that a full-fledge dissemination system should
be supported through satellite communication (Singh et al., 2016). This results in episodes such as cyclone
Ockhi wherein lack of timely alerts costs many lives (Roshan M, 2018). Such dissemination systems are to be
provided along with feedback mechanism, to validate the services in real or near-real time. The feedbacks also
open new avenues of data collection. Crowdsourcing of data gathering could play an essential role in making
seafarers eyes and ears of the scientists (Kelly et al., 2020).
**3. Recommendations**
All of the seven themes of the UNDOSSD have major implications to the needs of the IOR countries, and has
stakeholders at its heart. Hence, operational services can be one of the most important approaches for the
community engagement in the fulfilment of theme's goals. The existing frameworks of IIOE-2 and others (e.g.
RIMES) may be utilized to ensure that the IOR countries can share their know-how (e.g. digital
gene/metabarcoding bank) and have access to the operational services, which will fulfil 'transparent ocean' and
'inspiring and engaging ocean' goals. Reliable maps of oil-spill and marine debris are essential for meeting the
goal of 'clean ocean'. In addition to the adverse physical sea-state, algal bloom and jellyfish swarming
predictions are required for achieving 'safe ocean' objective. Coral-bleaching and SCUBA-buddy advisories can
form important pillar to 'healthy and resilient ocean' by supporting ecotourism-based alternate livelihood.
Telemetry, acoustics and genomics aided by satellite data and model simulations will enable us in managing
responsible fisheries through the virtue of 'predictive ocean'. This can in turn ensure 'sustainable and productive
oceans' for the coming generations.

**Acknowledgements**
The author appreciates motivation from the Director, INCOIS to take up this study. This publication was co-
sponsored by the Scientific Committee on Oceanic Research (SCOR) as part of the activities of the International
Indian Ocean Expedition 2. The author is grateful for the encouragement from the co-founders of the IIOE-2
ECSN (Early Career Scientists' Network). This is INCOIS contribution no. 418.

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

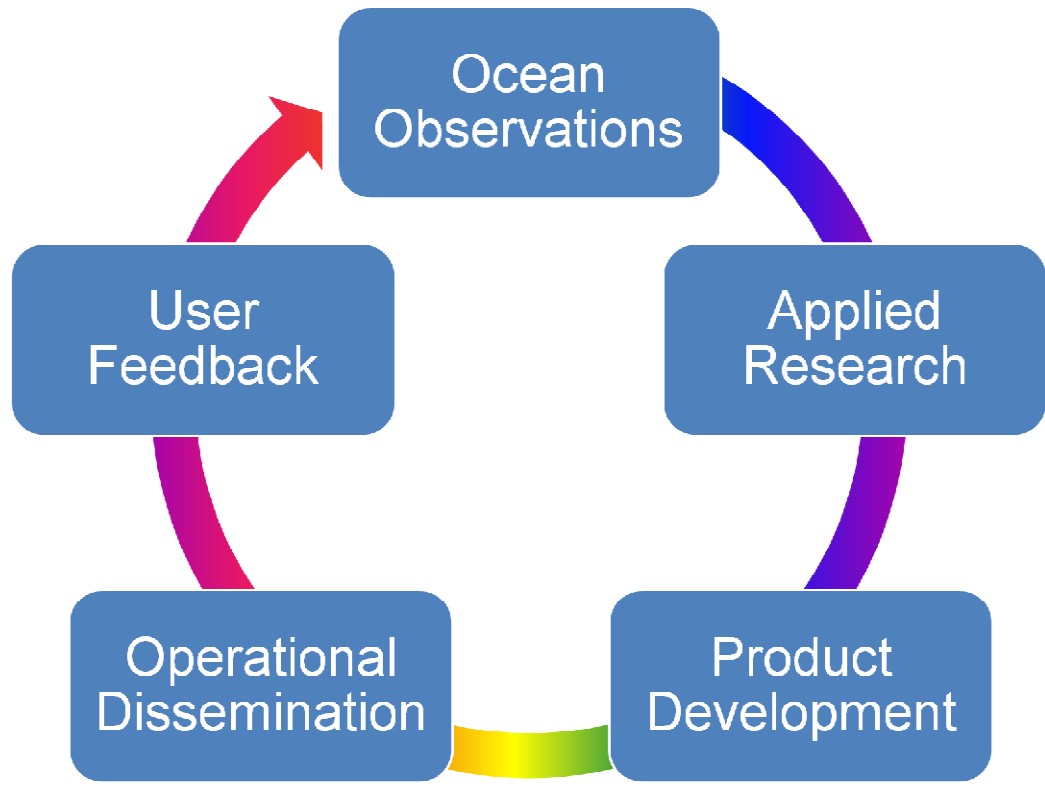


Fig.01. Evolution and life-cycle of an operational ocean service. After identification of user requirement and
consultation with them, ocean observations are planned or retrieved (if already existing). The observations are
then used for the research focused to the development of product customized for stakeholder requirement. The
products are then disseminated as routine operations, and feedbacks from the users are collected for further
improvement of the product. Due to the continuous advancement of the technology and change in the user
requirements, this cycle repeats to find the best achievable solutions for the contemporary needs at the user-
level.