# Peer review of "Ideas and perspectives: Ushering Indian Ocean into the UN"

_Biogeosciences, 2020_

## Author Comment (AC1)

**Response to bg-2020-492-RC1**

This is an interesting and timely piece by Kumar, who is an early career scientist. Overall, it is well written but some discussion should be strengthened. I list a few points below:

R: Thank you. The responses to your comments are in blue fonts, and marked with R.

1. I would suggest the author to summarize text in a couple of graphics/schematic figures. Readers may save their time while reading the article and get a better picture of author's take on the subject.

   R: Thank you for this valuable suggestion. A figure is permissible as per the journal's guideline on this category (Ideas and Perspectives) of submissions. Thus, a schematic is included in the revised manuscript.

2. In such a paper, one would like to see a "Recommendations" section. Within the UN decade frame work, the author should provide his own take how the decade's objectives specific to Indian Ocean would be achieved. If decade does not have Indian Ocean specific objectives, then would the author recommend to UN to also include some Indian Ocean specific objectives?

   R: Thank you. The decade objectives, though global, fit well with the requirements of the Indian Ocean science and its rim countries. A recommendation section, which also serves as the conclusion, is added in the revised manuscript.

3. This paper is going to part of special issue focusing on research progress made since the inception IIOE-2. But there is no mention of IIOE-2 in the manuscript. In the introduction section, author should have some comments on the ongoing IIOE-2 program.

   R: Thank you. The suggestion is very pertinent. The introduction section has been modified to highlight how multilateral initiatives such as IIOE have led to better understanding of the IO.

4. The Bay of Bengal Initiative for Multi-Sectoral Technical and Economic Cooperation (BIMSTEC) supports international trade among India, Bangladesh, Bhutan, Myanmar, Thailand, Sri Lanka, and Nepal. BIMSTEC should find a place in the manuscript.

   R: Thank you. The BIMSTEC has been included appropriately in the revised manuscript.

5. Acronyms have been overused. It is hard to memorize so many of these while one reads the paper. They also take the reading flow away. Unless an acronym is needed >10 times, it is of no use; and there is no need to use acronym for phrases like the "Arabian Sea" as AS even if it is used >10 times.

R: Thank you and apologies. The revised manuscript has reasonable usage of acronyms.

6. The author mentions Noctiluca blooms. This is major and relatively recent phenomenon in the Arabian Sea of whose impacts on the ecosystem are unknown. Author may go throw Rixen et al's paper in this special issue and enlarge the discussion on this. Reference: Rixen, T., Cowie, G., Gaye, B., Goes, J. I., Gomes, H., Hood, R. R., Lachkar, Z., Schmidt, H., Segschneider, J. and Singh, A.: Reviews and syntheses: Present, past, and future of the oxygen minimum zone in the northern Indian Ocean, Biogeosciences, 17, 1–30, https://doi.org/10.5194/bg-17-1-2020, 2020.

R: Thank you. This recent work has been cited appropriately in the revised manuscript.

7. There are some typos/ language/grammatical mistakes that should be corrected. For example- "north ocean" -> north in line 37

R: Thank you and apologies. A sincere attempt has been made to minimize such errors in the revised manuscript.

---

## Author Comment (AC2)

**Response to bg-2020-492-RC2**

This is a timely early career perspective that correctly notes both the importance of Marine Ecosystem Research and Operational Services in the Indian Ocean (IO) going forward, and the fact that such research and services took somewhat a back-seat compared to other basins in previous decades. Overall, it is well written. I would recommend that some discussion could be expended, either directly, or through additional references, and that some language improvements could be made:

R: Thank you. The responses to your comments are marked with R.

(1) In the introduction, the author correctly mention the recent shift to operational oceanography. This shift is international and it is worth mentioning or citing initiation of the first international operational satellite oceanography symposium in 2019 (with a second one planned for 2021), as an international global effort.

R: Thank you for this useful addition. The manuscript is revised to depict the global trend.

(2) The author correctly notes the great opportunities opened by the proliferation of satellite missions and the "swarm" of nano-satellites. This is correct, however, it comes with several challenges that should be equally discussed here, or at least mentioned by reference. These include problems in representation due to satellite orbit requirements (geostationary, polar orbiting) that include over and under spatial sampling, aliasing, unresolved/upsampled variability, as well as issues that arise from the deluge of satellites, to name a few. A recent short EOS news article highlighting the satellite overcrowding challenge is: Bruinsma, S., M. Fedrizzi, J. Yue, C. Siemes, and S. Lemmens (2021), Charting satellite courses in a crowded thermosphere, Eos, 102, https://doi.org/10.1029/2021EO153475. Published on 19 January 2021.

R: Thank you. Not only this recent work has been cited in the revised manuscript but the alarm/alert has been extended to the ocean observations, that these platforms may add to the marine debris after its lifespan.

(3) The author correctly highlights the importance of satellite communication based alerts during fair weather. The same is true and perhaps even more crucial during storms, and should be discussed.

R: Thank you. Though Ockhi cyclone tragedy due to lack of satellite communication has been mentioned, the section is modified to bring out the message in a clearer way.

(4) There are numerous typos/ language/grammatical/auto-speller mistakes, that should be corrected. A list was provided directly to the author as it is not relevant for the discussion.

R: Thank you and apologies. The list is well-received and mistakes are corrected. Further, a sincere attempt has been made to minimize the errors in the revised manuscript.

---

## Author Comment (AC3)

**Response to bg-2020-492-CC1**

The ideas and perspectives given by Nimit have potential with important implications for the sustainable use of the Indian Ocean's resources. However, we strongly advise that some major revisions are made before the article can be considered for publication. Significant changes can be made to improve the structure of the article and to add details to the arguments, which in some cases are lacking the appropriate level of information. These adjustments would reinforce the findings that are already given in the article but are not currently explicit enough. Further, we find some minor adjustments, mostly related to spelling and grammar that will improve the communication of the paper.

R: Thank you. The responses to your comments are in blue fonts, and marked with R.

**Major Criticisms**

**1. Structure**

Overall, we find that the structure of the article lacks signposting and there is only a limited use of subtitles. Therefore, it is difficult for the reader to follow the key arguments. Additional subtitles would organise the information more clearly, defining all the important information.

For example, the main body of work entitled "Marine Ecosystem Research and operational Services (MERS)" could be divided into smaller sections to divide up key arguments. Further, there is no conclusion, which is essential to collate each of these arguments into an overarching finding.

Each of these sections could then benefit from introductory topic sentences to highlight the key message being presented.

R: Thank you. The submissions in this category (Ideas and Perspectives) are expected to be short, as per the journal guidelines. Nevertheless, the possibility to reasonably provide subsections will be explored in the revised manuscript.

**2. Definitions**

Before addressing specific sections of the article, an important criticism is that 'operational' or 'operating' services are not clearly defined for the reader. It is essential to give a clear explanation of this key term in this specific context.

R: Thank you. This will be made clearer in the introduction section.

**3. Abstract**

Whilst the abstract does outline some key points made in the paper, such as the need to consider more operational services like Coral Bleaching alerts and SCUBA-assisted advisories, as opposed to

only fishery information, it does not include key information that we would usually expect from an Abstract. Firstly, the abstract states that the paper will be a review. However, the main results or concluding arguments from such a review are not presented. The purpose and aims of the paper should be more explicit and subsequently the results to these aims must also be made clear. Secondly, whilst some background information is given, the abstract does not allude to the rationale of the paper and the importance of the findings, which we would argue are fundamental to supporting the perspective being presented. A clearer progression from background information, purpose and review to a conclusion would clarify why new operational services should be considered. Thirdly, a handful of examples of issues are presented in the abstract, the significance of the selected examples are unjustified and it is unclear on reading the paper what the major issues and minor issues are.

R: Thank you for pointing out the word 'review' that created the confusion. The same will be avoided in the revised manuscript. The manuscript type is 'Ideas and Perspectives' and the abstract has been prepared to suit the same.

**4. Introduction**

The introduction contains a lot of important information. Unfortunately, the significance of these statements are somewhat lost due to poor structure. Like the abstract, improving the structure of the Introduction would give the arguments presented in this perspective a more explicit motive. Specifically, we would recommend that the introduction follows an inverted pyramid structure; starting with widely applicable background and contextual information and ending with more specific and narrowly targeted points. This structure will guide the argument and emphasize the importance of this perspective and inform the reader of the scope, which we also find should be more explicit. For example, the introduction already begins with the wider context of the Indian Ocean, mentioning population growth, highlighting how many lives depend on the resources of the IO and addressing the multiple cultures that inhabit IOR countries. Further detail could be given here as it would be interesting to explore the specific differences between the populations that inhabit IOR countries and use the IO. In addition, more detail should be given to the specific threats that target marine ecosystems, to highlight the specific issues that need to be overcome for the sustainable and equal use of the IO. Together, these added details would enable the reader to understand to what extent impacts in the IO affect people's everyday lives and therefore the importance of the perspective.

The introduction could also include introductory information about the current environmental and ecological conditions of the IO. Whilst the introduction states that the IO is poorly studied and there is little information available, existing literature should still be outlined. For example, details on other uses of the IO rather than fishing like the extent of tourism. Or published information on the threats specific to the IO like evidence of coral bleaching events. Perhaps fundamental facts like the climatic cycles, weather and temperature changes, salinity or nutrient information, to highlight both the physical as well as a social factors that make the IO unique from other oceans. Again, this information would highlight why advocating sustainable use of the IO is so important.

If the preceding information is considered, this final recommendation may not be necessary as they will be implied: we suggest explicitly outlining your research question or purpose.

Finally, the hypotheses state that "the best possible approach to win the community attention (and to invoke the responsible ownership of resources) is to engage the communities with the operational ecosystem related services" and it is unclear how the rest of the article proves or disproves this hypothesis. Although there are a few mentions of how communities can get involved with operational services, a separate paragraph or section to explain the outcomes for this hypothesis would clarify the findings.

Usually we would expect to see a brief section to summarise the methodology next. This does not need to be extensive but as this paper explicitly states that it is a review, how the review was conducted needs acknowledgement. For example, how were the papers selected for review? Was it systematic?

R: As aforementioned, apologies for the confusion created due to the word review. This manuscript follows the guidelines of an 'Ideas and Perspectives' type of submission, abiding to which attempt will be made to incorporate your suggestions in the revision.

**5. Marine Ecosystem Research and operational Services (MERS)**

The main section of the work could be improved by adding more details and relevant information to the arguments. For example, the use of satellites is covered in depth, however there is little information to explore how they can or have been used as an operational service. Developing the application of these tools is more appropriate for the hypothesis in question. Similarly, the main section of work tends to focus on technologies and problems for the Indian Ocean. Instead, the main focus should be on specific operational services and why they are important, followed by what kind of data/ technology is required to put this operational service into action. Simply put - your recommendations. For example, rather than focussing on the problem of plastic pollution, the focus should be on the operational service of plastic detection, why it is necessary and how it can be achieved.

A critique is developed against the lack of information available for the Indian Ocean. To shine light on the areas of information that could be helpful for the sustainable use of the IO's resources, it would be interesting to detail the kind of information that has been useful in other ocean basins for operational services.

In general, more case studies and greater referencing would elevate the article as a whole.

R: Thank you. These topics will be dwelled in detail as much permissible for this article type.

**6. Use of a Figure**

At present, this article does not contain any figures or imagery. However, a figure that synthesises the problem (sustainable future use of the IO) alongside the solutions (use of various operational services) could be very beneficial to summarise all of the ideas and highlight the core ideas being presented.

R: Thank you. This has been also recommended by an anonymous reviewer and a summary figure shall be incorporated in the revision.

**Minor Criticisms**

There are several adjustments that could be made to facilitate legibility and to improve the important messages within the piece.

Firstly, we do not find it necessary to include "An early career's take" in the title as it is not relevant to the findings in the paper.

Secondly, we think that both the introduction and main body of text could benefit from the support of additional references. For example, in the introduction the statements on population pyramids profiles of IOR countries and IOR marine ecosystems under threat statement should be qualified by references.

Alongside these adjustments, we find several grammar issues that should be remediated, which are listed below. We also suggest using a grammar checker to find any further errors.

| Line | Mistake | Correction |
|---|---|---|
| 30 | "Majority of the IOR countries have either expansive *of* stationary population pyramid profile, with limited land resources" | Replace 'of' with 'or' |
| 34 | "consumerism is poised to extend to larger *part* of the population" | Replace 'part' with 'parts' |
| 36-37 | "It is thus important to ensure that the resources in this unique, land-locked from the north ocean, are harnessed in harmony with the sustainability and with the equal opportunities" | Needs rewording, for example:

"In this unique and land-locked from the north ocean, it is important to insure that the resources are harnessed sustainably and there is equality of opportunity" |
| 45 | "This paper *hypothesize*" | "hypothesizes" |
| 82 | "With more than two *decade* of uninterrupted observations" | "decades" |
| 87 | "The research can then support to the operational services" | Remove "to" |
| 112-113 | "Human activities can be nuisance to the marine environment in many ways and sound is one of the ways" | Could be worded better, for example: Many human activities can be a nuisance to the marine environment and can cause impact in several ways, such as through sound |
| 122 | "Plastic adversely *impact* the marine…" | "impacts" |

R: Thank you. The revision includes additional references and corrections of grammar/language as suggested by you as well as reviewers.

---

## Author Response (AR1)

Kumar Nimit,

IIOE-2 ECS,

INCOIS.

Date: 10 May, 2021.

Associate Editor

IIOE-2 inter-journal SI.

Dear Sir,

Sub.: Submission of minor revision of manuscript no. bg-2020-492.

With above reference, I would like to hereby submit the minor revision of the manuscript titled 'Ideas and perspectives: Ushering Indian Ocean into the UN Decade of Ocean Science for Sustainable Development (UNDOSSD) through Marine Ecosystem Research and Operational Services – An early-career's take'. I would like to convey that I have addressed all the comments by two anonymous reviewers as well as a community commenter. I believe that this manuscript will help make significant contribution in the IIOE-2 inter-journal special issue.

Yours Sincerely,

Kumar Nimit.